# Fungistatic and Bactericidal Activity of Hydroalcoholic Extracts of Root of *Jatropha dioica* Sessé

**DOI:** 10.3390/microorganisms13051027

**Published:** 2025-04-29

**Authors:** Lizeth Aguilar-Galaviz, Jorge Cadena-Iñiguez, Dalia Abigail García-Flores, Gerardo Loera-Alvarado, Diego Rivera-Escareño, María Azucena Ortega-Amaro

**Affiliations:** 1Colegio de Postgraduados, Innovación en Manejo de Recursos Naturales, Campus San Luis Potosí, Iturbide 73, Salinas de Hidalgo 78600, SLP, Mexico; aguilar.lizeth@colpos.mx (L.A.-G.); garcia.dalia@colpos.mx (D.A.G.-F.); gerardo.loera@colpos.mx (G.L.-A.); diego.rivera.e97@gmail.com (D.R.-E.); 2Coordinación Académica Región Altiplano Oeste, Universidad Autónoma de San Luis Potosí, Carretera Salinas-Santo Domingo 200, Salinas de Hidalgo 78600, SLP, Mexico; azucena.ortega@uaslp.mx

**Keywords:** revalorization, biological control, secondary metabolites, antimicrobial, postharvest

## Abstract

*Jatropha dioica* Sessé (JD) is a plant from arid and semiarid zones of Mexico related to local therapeutic uses and possible use in food and agriculture as a control agent of pest organisms that helps to reduce impacts on the environment, human health and resistance by phytopathogens. In vitro bactericidal activity was evaluated with the well diffusion method in doses of 1000, 2500, 5000, 7500, 10,000 and 20,000 µg mL^−1^, and fungistatic activity was evaluated with the agar dilution method (500, 1000, 1500, 2000 and 4000 µg mL^−1^) in *Pseudomonas syringae*, *Botrytis cinerea* and *Fusarium oxysporum* using hydroalcoholic extracts of *J. dioica* root in a completely randomized design with five replications. Total phenol and flavonoid contents were recorded by the Folin–Ciocalteu and aluminum chloride methods. Ethanol and methanol extracts showed fungistatic activity on *B. cinerea*, inhibiting from 42.27 ± 1.09 to 46.68 ± 0.98 mg mL^−1^, with an IC_50_ of 5.04 mg mL^−1^, with no differences by solvent type. In *F. oxysporum*, inhibition ranged from 14.77 ± 1.08 to 29.19 ± 0.89 mg mL^−1^, and the methanol extract was more efficient, generating a stress response to the ethanol extract. The bactericidal activity on *P. syringae* recorded inhibition zones of 17.66 ± 0.33 and 16.66 ± 0.33 mg mL^−1^, with ethanol being more efficient. The phenol content ranged from 8.92 ± 0.25 to 12.10 ± 0.34 mg EAG g^−1^ and flavonoid content ranged from 20.49 ± 0.33 to 28.21 ± 0.73 mg QE g^−1^ of sample dry weight. The results highlight the biological activity of *J. dioica* as an alternative to biopesticides that minimize agrochemical applications and generate pathogen resistance. These advances contribute to the revaluation and conservation of the species.

## 1. Introduction

The arid, semiarid and hyperarid zones in Mexico constitute 63% of its surface [1], and its rural population depends on non-timber forest resources (NTFRs) [2,3]. Some are the nopal (*Opuntia ficus-indica*), yuca (*Yucca* sp.), gobernadora (*Larrea tridentata*) and sangre de drago (*Jatropha dioica Sessé*), among others [4], and as a response to environmental limitations, many of these species synthesize a wide range of secondary metabolites (Sm), which have recorded biological activity oriented to public health or antimicrobial applications [5]. Through bioprospective research, metabolites can be used for the development of products with economic value and generate rural projects in the medium term [6]. Such is the case of *J. dioca* (Euphorbiaceae), an RFNM of microphyllous desert scrub vegetation growing in calcaric regosol soil, at altitudes ranging from 1851 to 2100 [7], from whose biomass flavonoids, lactones, quinones and sterols have been recorded [8,9].

Currently, plant scientists have focused on designing applications for the control of pest organisms in both crops and harvested products. Normally, high-impact synthetic pesticides are applied that induce negative effects on the environment (water, air, soil) and biodiversity as well as cause health problems, such as cancer, diabetes, reproductive, respiratory and neurological disorders [10,11]. An alternative is to derive low-cost economic and environmental benefits from plant extracts that reduce or eliminate pathogenic microorganisms [11,12,13]. Economic losses in agriculture and post-harvest life due to pest organisms [14] of 40% [15,16] have been recorded worldwide, which put food security at risk [17,18].

The fungus *Fusarium oxysporum* impairs the absorption of water and nutrients by altering the metabolism of the plant, inducing chlorosis, wilting, flower shedding and vascular necrosis [19,20,21]. *Botrytis cinerea* is the causal agent of gray mold [22], a necrotrophic pathogen that causes losses in more than 500 species of plants in the field and storage, attacking stems, leaves, flowers, fruits and seeds with impacts of up to USD 100,000 million [23,24]. Similarly, the bacterium *Pseudomona syringae*, causal agent of bacterial canker (aerobic Gram-negative), has associated losses in more than 40 plants annually, with up to 25%, and it survives as a saprophyte [25,26].

To prolong the useful life of crops, the FAO (2024) suggests continuous monitoring and being able to respond quickly with environmentally sustainable preventive control strategies [18]. Chemical treatments are regulated and restricted by their associated impacts [27,28,29,30]. The use of formulations based on biological extracts can be an ally in reducing pesticide use and avoiding resistance that requires the use of larger quantities or the aging of molecules. Based on the above, hydroalcoholic extracts of *Jatropha dioica* Sessé roots were evaluated on the in vitro growth of *Botrytis cinerea*, *Fusarium oxysporum,* and *Pseudomonas syringae*, with the aim of designing formulations and compositions for agricultural use and postharvest handling in the medium term.

## 2. Materials and Methods

### 2.1. Collection of Plant Material and Preparation of the Extract

*Jatropha dioica* (Euphorbiaceae) was collected during the dry season, January–April, 2024 in Ejido Loma de la Carreta, Villa González Ortega, Zacatecas, Mexico (23°11′ and 22°27′ N and 101°22′ and 101°57′ W), with a mean annual temperature and precipitation of 19 °C and 373.9 mm [31], respectively. First, 602 g of root was collected, chopped and dried at 20 °C ± 2 and 60% relative humidity without an oven for 7 d. The particle size was made uniform with a HC-700Y mill (Grinder, Zhengzhou, Henan, China) and a No. 20 sieve. To obtain the extracts, 50 g of dry sample was macerated with 500 mL of 80% ethanol for 7 d, making three solvent changes every 24 h. Similarly, 50 g of dry and ground root was macerated with 250 mL of 99.9% solvent for 7 d. They were then vacuum-filtered with filter paper and concentrated in a rotary evaporator (RE100-PRO, DLAB, Los Angeles, CA 90015, USA) at 120 rpm at 40 °C. They were placed in a 9023A drying oven (ECOSHEL, PHARR, TX, 78577, USA) at 40 °C for one week and stored in refrigeration until use [32,33]. The yield of the hydroalcoholic extracts was calculated through the following equation:

Yield (%) = (W1 × 100)/W2, where W1 is the dry weight of the extract after rotary evaporation and W2 is the dry weight of the plant used, recording 10.32% for ethanol and 1056% for methanol.

### 2.2. Inoculum Preparation

The inocula were provided by the laboratories of the National Institute of Forestry, Agriculture and Livestock Research and the Potosino Institute of Scientific and Technological Research. The *B. cinerea* strain B05.10 was cultured on potato dextrose agar (PDA) medium and incubated at 25 °C for four days before use. The *Pseudomonas syringae* DC3000 strain was replated onto Luria-Bertani (LB) agar and incubated at 37 °C for 48 h before use.

The fungal strains *B. cinerea* and *F. oxysporum* were reseeded after 4 and 7 days of growth. A 4.6 mm portion of mycelium was aseptically plated on potato dextrose agar (PDA) for evaluation. In the case of the *P. syringae* bacteria, it was inoculated in Luria–Bertani (LB) medium. After incubation, a colony was taken with a bacteriological loop and aseptically transferred in saline water (0.85%). For 5 min, the suspension was diluted in a vortex followed by a suspension adjustment in a Genesys 10S Vis spectrophotometer (Thermo Fisher Scientific, Lenexa, KS, USA) at 625 nm based on the 0.5 McFarland turbidity standard (106 CFU mL^−1^) [33].

### 2.3. Preparation of Stock Solution and Fungistatic Activity

To obtain the stock solution with a concentration of 8666.66 µg mL^−1^ of hydroalcoholic extracts, 0.13 g was weighed in 15 mL of distilled water in falcon tubes (50 mL) and placed in an ultrasonic bath for complete dilution. For the fungi, the agar dilution method was used in the potato dextrose agar (PDA) culture medium from DIBICO. In the case of the bacteria, the susceptibility method was the well diffusion method in Luria–Bertani (LB) agar [34]. The PDA medium was added considering the concentrations, repetitions and type of extract to be used (5 × 5 × 1), and the result was multiplied by the volume needed for a 90 × 15 mm Petri dish (20 mL) and divided by two to obtain a 50:50 ratio. Therein, 50% was concentrated PDA medium and 50% was water and extract. When preparing the PDA medium, the result (g) was doubled, as was the extract concentration (mL). The mixture of the extract with the water of the culture medium was considered, since the concentration should not be affected by the water contained in the PDA medium. From the mother solution of the hydroalcoholic extracts, the concentrations were prepared with a final volume of 50 mL (Table 1).

The concentrations were added to the culture medium, sterilized in an autoclave (121 °C for 15 min) and added to Petri dishes for solidification, and a 4.6 mm piece of mycelium incubated at 25 ± 2 °C was placed in the center. Measurements were taken every 24 h for five days or until the control reached its full growth [28]. Finally, after the incubation time, the growth inhibition halo was measured. The results were obtained as an average ± standard error, and the percentage of inhibition was determined through the following formula:% of mycelial growth inhibition = diameter (mm) of the negative control—Diameter (mm) of the phytopathogen growth in the extract/diameter (mm) of the negative control × 100.

Through a linear regression in RStudio-2024.12.1-563, the IC_50_ (minimum dose that achieves a 50% decrease in mycelial growth) of the hydroalcoholic extracts was estimated. The percentage of growth inhibition of the hydroalcoholic extracts of *J. dioica* root was obtained through the measurement in millimeters (mm) for five days of the two phytopathogenic fungi (*Botrytis cinerea* and *Fusarium oxysporum*). A positive control was considered using the concentration provided by the manufacturer of a commercial fungicide called Ridomil Gold (metalaxyl-M: methyl N-(2,6-dimethyl phenyl)-N-(2-methoxy-acetyl)-D-alaninate at 4%, and 64% mancozeb: ethylene bisdithiocarbamate of manganese and zinc) and a negative control. A repeated-measures ANOVA was performed in STATISTIC 7.0 with Tukey tests (*p* ≤ 0.05).

### 2.4. Bactericidal Activity

In the case of the bacteria, the susceptibility method was the diffusion of wells in *Luria–Bertani* (LB) agar [35]. The stock solutions of the hydroalcoholic extracts were prepared, 50 mg were weighed in 1000 µL of distilled water dissolved in a sonicator bath for 5 min, from which the following concentrations were obtained (Table 2).

A 10 µL aliquot of the adjusted suspension was applied to the LB medium. This was inoculated into the corresponding media with a uniformized Drigalski loop and left to stand for 5 min. Then, wells were made with a sterile 6 mm punch, followed by the application of 100 µL of the different concentrations with five repetitions, which were left to stand for one hour and were incubated at 37 °C for 18 h [32]. Then, the inhibition zone was measured in millimeters with a vernier. A one-way ANOVA was performed and the Tukey post hoc test was run (*p* ≤ 0.05) in STATISTIC version 7.0.

### 2.5. Total Phenol Analysis

An amount of 0.3 mL of each extract concentration (0.50–4 mg mL^−1^) was added to test tubes, followed by 1.5 mL of Folin’s reagent (10%) and 1.2 mL of Na_2_CO_3_ (7%). The mixtures were shaken and incubated for 30 min at 40 °C in a water bath. Finally, the absorbance at 760 nm was measured, using 1.5 mL of Folin’s reagent and 1.2 mL of Na_2_CO_3_ (7%) as a blank. The samples were prepared in triplicate and the averages were used to fit a calibration curve with gallic acid [36].

### 2.6. Preparation of Gallic Acid Standard and Analysis of Flavonoids

A standard solution of gallic acid was prepared by dissolving 10 mg in 10 mL of distilled water (1 mg mL^−1^), from which various concentrations were prepared in 10 mL volumetric flasks (9.4 × 10^−3^−1.5 × 10^−1^ mg mL^−1^) [35]. Flavonoid content was determined by the colorimetric aluminum chloride assay. Then, 0.5 mL of each extract concentration was added to a test tube containing 2 mL of distilled water. At the same time, 0.15 mL of 5% NaNO_2_ was added and, after 5 min, 0.15 mL of 10% AlCl3 was added. After 6 min, 2 mL of 1 M NaOH was added to the mixture. The volume of the mixture was brought to 5 mL by immediately adding 1.2 mL of distilled water.

The absorbance was measured in a spectrophotometer at 510 nm. Readings were taken in triplicate and the average absorbance value was used to calculate the total flavonoid content. The flavonoid content was expressed as quercetin equivalent (mg QE/g) using the linear equation based on the standard calibration curve [36]. Regarding the preparation of the quercetin standard, 20 mg of quercetin was weighed and diluted in 10 mL of distilled water, with which the concentrations for the calibration curve were prepared (0.0625–4 mg mL^−1^).

## 3. Results

### 3.1. Total Content of Phenols and Flavonoids

The concentration of polyphenolic compounds (phenolic acids) and flavonoids obtained in the hydroalcoholic extracts is shown in Table 3. The total phenol content was determined using the Folin–Ciocalteu method. A calibration curve was created with the gallic acid standard, from which an equation (y = 9.5597×; R^2^ = 0.9956) was derived, expressed in milligrams of gallic acid equivalents in grams of dry weight of the sample (mg EAG g^−1^). Similarly, the total flavonoid content in the hydroalcoholic extracts was calculated using a calibration curve equation (y = 0.3718×; R^2^ = 0.9993) and expressed in milligrams of gallic acid equivalents in grams of dry weight of the sample (mg EQE g^−1^).

### 3.2. Fungistatic Activity

During the five days evaluated, there were no significant differences between the negative control and the treatments. However, after 24 and 48 h (Figure 1a,b), significant differences were recorded as the positive control was the same at all doses evaluated, and at 72 h (Figure 1c), the positive control did not record differences with doses of 500 and 1000 µg mL^−1^. For both extracts after 96 and 120 h (Figure 1d,e), significant differences were observed from 1500, 2000 and 4000 µg mL^−1^, the latter being the concentration that registered the highest inhibitory value with 42.27 ± 1.09% and 46.68 ± 0.98% and a calculated IC_50_ of 5482.21 and 5024.35 µg mL^−1^ for ethanol and methanol, respectively. No significant difference was observed between the use of solvents for extraction, even though methanol is of greater polarity and metabolite drag.

Figure 2 indicates after 120 h, *F. oxysporum* recorded an effect on the ethanol extract at 1000 and 2000 µg mL^−1^, generating greater growth compared to the negative control. Regarding the positive control, there were no significant differences with the ethanolic doses; however, with methanol, differences were observed from 1000–4000 µg mL^−1^. The highest dose of ethanol 4000 µg mL^−1^ (14.77 ± 1.08) showed greater inhibition; however, when compared to the positive control, there was no statistical difference. When comparing the 4000 µg mL^−1^ dose of ethanol and methanol, significant differences were observed between solvents, with methanol achieving greater inhibition (29.19 ± 0.89).

### 3.3. Bactericidal Activity

The mean growth inhibition halo of *P. syringae* treated with hydroalcoholic extracts of *J. dioica* root was obtained at 18 h, and according to the analysis, the doses of ethanol and methanol at 1000 µg mL^−1^ were equal (Figure 3).

Figure 3 shows that the ethanolic extract limited the growth of *P. syringae* to a greater extent; in some cases, the ethanol concentration of 2500 µg mL^−1^ was statistically similar to methanol concentrations of 2500, 5000, 7500 and 1000 µg mL^−1^. This showed that ethanolic extraction was more efficient (*p* ≤ 0.05) in controlling the growth of the *P. syringae* bacteria with a mean inhibition halo of 16.66 ± 0.33 and 17.66 ± 0.33.

## 4. Discussion

The presence of phenols and flavonoids in plant extracts helps corroborate the results we obtained, since biological activities must be attributed to a set of metabolites or a single specific compound. This has been observed in previous studies, which link fungicidal, bactericidal, antioxidant, and other activities to phenols and flavonoids. Since some treatments with gallic and pyrogallic acid reduced the development of early blight symptoms caused by *Alternaria solani*, the total or partial use of pesticides can be reduced [37]. Also, a mixture of phenolic compounds has high activity in cereal crops and is reported to be correlated with fungicidal and bactericidal activity [38]. Similarly, *Spergularia marina* is correlated with antioxidant activity (r = 0.909, *p* ≤ 0.01) and protocatechuic acid (r = 0.859, *p* ≤ 0.01) [39].

The antifungal activity of phenolic compounds, flavonoids, or other groups can be assessed through molecular descriptors, which unify the topological and electrostatic properties of the molecules’ frontier orbitals. Therefore, through the electrophilicity index (ω), phenolic compounds with greater antifungal activity have a low index, in contrast to those without activity, which has a higher index [40]. In other cases, flavonoids are important in the relationship between defense mechanisms and enzymatic activities that exist in plants against infections; hesperidin and naringenin are identified [41].

Different concentrations of total phenols, flavonoids and terpenes are reported in different organs of the plant (root and stem), with higher contents in the root regardless of the extraction method and solvent [42]. In the stem, the drying process was carried out through lyophilization, sterilized and non-sterilized extracts were evaluated with values of 194.31 ± 7.04 and 99.25 ± 2.50 mg EAG g^−1^, and the dry weight of the sample was considered [43]. The obtained values are higher than those obtained in the present study and could be due to the lyophilization used since in some reports lyophilization increases the amount of compounds [44], but it is a costly process that involves time and an adequate application of temperature so as not to intervene in the final properties of the product [45]. However, extraction with hydroalcoholic solvents demonstrates that the compounds present in the plant are thermostable, as they are maintained upon exposure to high temperatures [46].

The phenol content obtained from the ethanol and methanol extract of *J. dioica* was 8.92 ± 0.25 and 12.10 ± 0.34 mg EAG g^−1^ (Table 3). These data, when compared with the third and fourth studies, double the amount of total phenols in the root. The total was 46.06 ± 4.14 mg GAE g^−1^ of extract [47], in which maceration and the hydroalcoholic solvent were used as the extraction methods. Similarly, ref. [48] evaluated the presence of total phenols using two extraction methods: heat reflux (2.34 ± 0.93 mg g^−1^) and microwave-assisted extraction (1.80 ± 0.48 mg g^−1^) per gram of biomass. These data are lower than those obtained in the present study, and the difference could be due to the extraction method and solvent used, since the effect of temperature, particle size, time and solvent use influences the yield, stability and content of bioactive compounds.

Extraction methods are currently classified as conventional and modern, whose main function is to optimize the process by using fewer solvents and obtaining higher yields. However, some have disadvantages, as some techniques that use heat decompose or degrade plant tissue into thermolabile secondary metabolites. However, it would be important to optimize extraction processes for the plant tissue of interest, since each sieve is different. All possible variables were controlled to obtain higher yields of compounds with economic viability, less time and energy expenditure and improved yields without affecting biological activity [49,50,51].

The flavonoid content obtained from the *J. dioica* extract with ethanol and methanol was 20.49 ± 0.33 and 28.21 ± 0.73 mg QE g^−1^. In accordance with studies reporting flavonoid content in *J. dioica*, both pieces of data are different from those obtained in the present study, for example, 36.16 ± 1.5 QE µg mL^−1^ by [49] and 2.25 ± 0.10 mg QE g^−1^ by [47]. This may be due to various climatic factors, such as variation in altitude, soil type, climate, phenological stage of the species, part of the plant used and some genetic factors [52,53,54].

As observed in Table 3, the flavonoid content (20.49 ± 0.33 and 28.21 ± 0.73 mg QE g^−1^) is higher than total phenols (8.92 ± 0.25 and 12.10 ± 0.34 mg EAG g^−1^). This may be due to an analytical limitation in the methodology we used, since the absorption spectra and chemical structure of each molecule are different, and their use cannot be considered selective and specific [55]. However, some studies report higher quantification of flavonoids than total phenols, but this behavior is not disputed [56]. Also, the aqueous extract of *Thalictrum foliolosum* collected from different locations at different elevations obtained the same behavior with higher flavonoids than total phenols [53].

*Dillenia pentagyna* recorded 0.75 ± 0.03 mgGAE g^−1^ of phenols in seeds and 24 mg RUE g^−1^ of flavonoids [57]. The root of *Polyscias fruticosa* obtained a value of total phenols of 8.57 µgGAE mg^−1^ and flavonoids of 11.79 µgQE mg^−1^ [58]. Because there are antecedents of this behavior, it is considered that experimental factors intervene in the measurements of absorbance and its quantification, such as the amount of solvent used in the preparation or dilution of the samples, the standard that is used, the concentration of the alkaline medium, absorbance, temperature and reaction time—methodological aspects that underestimate the quantification of flavonoids [59]. Even some reagents used in these methodologies can react with other oxidizing molecules; inorganic, aromatic and aliphatic compounds; sugars; proteins or polysaccharides [60].

The flavonoids quercetagetin 7-O-glucoside and quercetagetin 7-O-arbinosyl galactoside, evaluated on the tested Gram-negative strains, were more sensitive, and differences were detected in the action of the compounds on RecA (−) mutant strains of *E. coli* and RecA (+) strains with an SOS repair system [61]. Similarly, aqueous, alcoholic, and ether extracts of *Simira cordifolia* against *Candida albicans* and *Escherichia coli* revealed the presence of saponins, flavonoids, phenols, tannins, amino acids and alkaloids, with broader results for the aqueous extract. The antimicrobial assay showed significant inhibition zones for both microorganisms, with E. coli being more sensitive, exhibiting zones of up to 20 mm, compared to *C. albicans* with zones of up to 18 mm [62]. In conclusion, as mentioned previously, variations may be due to the amount of sample and solvent used in the extraction of the compounds [58] and to climatic factors that contribute to the generation of different secondary metabolites.

There is great concern about the application of agrochemicals to crops, as it causes residues in the environment and food and even generates resistance in phytopathogens [63]. To minimize risks, it is necessary to implement management strategies and agronomic practices, such as sanitation measures, reducing fertilization, crop rotation, specific use of agrochemicals, analysis of nursery plants, protected crops and the use of biological control [64,65].

The generated resistance can vary depending on location, species and used chemicals [64], and resistance can be assessed by PCR, through the presence of indicators such as enzymes or genes [66]. Constant monitoring is necessary to identify resistance over time [67,68]. Recently, the combination of chemicals with biological control agents may be an alternative to minimize pesticide doses in the field [69]. Or, if applicable, the use of plant extracts can control phytopathogens in the field. Pathogens develop rapid resistance in response to multiple disease cycles per season. To ensure crop yields, pesticide applications are excessive and repeated. However, this action generates increasing resistance in pathogens [65].

For example, *B. cinerea* has developed resistance to several groups of fungicides: carbamates, quinone external inhibitors, dicarboximides, succinate dehydrogenase inhibitors, anilinopyrimidines and phenylpyrroles [64]. It also reports resistance to Boscalid (Cantus 50 WG), Fluopyram (Velum Prime 40 SC), Pyraclostrobin (Insignia 20 WG) and Fenhexamide (Teldor 50WG). These are succinate dehydrogenase inhibitor (SDHI) and quinone external inhibitor (QoI) fungicides, in which isolates with double resistance to several mutations were found, regardless of the geographical location, in sdhB (H272R, N230I and P225F/H) and cytb G143A (Malandrakis et al., 2022) [70]. Finally, a high percentage of resistance was also obtained in the β-tubulin gene mutation due to the use of carbendazim and methyliophanate (EC_50_ > 100 μg mL^−1^) [67].

Previous studies evaluated the growth inhibition of *B. cinerea* by hexane, dichloromethane, methanol and aqueous extracts of *V. amigdalina*, in which the dichloromethane extract had the highest percentage of inhibition (74.85–75.7%), followed by methanol, aqueous and hexane. However, the doses evaluated were 100–500 mg mL^−1^, doses higher than those used in our study, observing that 50% inhibition was obtained with 500 mg mL^−1^ with methanol, while 100 mg mL^−1^ registered only 30% [71]. Also, the extraction of polymeric proanthocyanidins from grape seeds on *B. cinerea* has been reported, in which the inhibition of mycelial growth increased progressively with increasing concentration, and the doses evaluated were 1 to 21 mg mL^−1^. An IC_50_ of 11.23–12.15 mg mL^−1^ was obtained, with the effect starting at 8 mg mL^−1^ [72]. Comparing these data with the present study, lower inhibition was obtained (46.68 ± 0.98 and 42.27 ± 1.09), with an IC_50_ of 5.04 mg mL^−1^.

Other studies evaluated the inhibitory activity of bay leaf extracts (*Laurus nobilis*) on *B. cinerea* isolates (2600–4200 µL L^−1^), recording that the highest dose had no fungistatic effect [73]. Other plants such as *Lomicera japonica* and *Bacharis trimera* had growth inhibition percentages lower than or equal to the present study, with values of 10% and 48%. However, several solvents were used to obtain an acid extract, and the obtained yields were not effective in inhibiting the growth of phytopathogens in the field [74]. A purified flavonoid called gnafalin A was obtained from *Pseudognaphalium robustuma*, which exhibited fungicidal activity against *B. cinerea* with an ED50 of 45.5 µg mL−1 [75].

Similarly, a 5% *w*/*v* concentration of ethyl acetate (EtOAc) extract from *Curcuma aromatica* generated 67.44% inhibition, as well as *Garcinia indica*, which recorded 79%. Other plants such as *Sechium compositum* have shown inhibition of *B. cinerea* growth using juice at low concentrations (1.0, 2.5 and 5%), with values of 83.15, 90.0 and 94.87% attributed to tetracyclic triterpenes [76]. Finally, a polyphenol-rich *Mangifera indica* leaf extract, microencapsulated by spray-drying to preserve its in vivo antifungal activity, demonstrated in vivo antifungal activity against *Penicillium digitatum* in oranges and *B. cinerea* in blueberries [77].

Depending on the climatic conditions where the plant is collected and even the type of extraction and solvent used can alter the presence and preservation of secondary metabolites [46]. In particular, *J. dioica* is distributed in arid and semiarid areas, where drought and salinity alter the metabolic and physiological processes of plants, which stimulates the production of terpenes, alkaloids, anthocyanins and others [78]. Also, the purity, structural stability and yield of secondary metabolites depend on the extraction method and solvent used.

Maceration is an economical and efficient extraction method for extracting thermolabile compounds such as polyphenols [79], and together, secondary metabolites are responsible for the fungistatic activity since they interfere with the vital process of fungi, changing the physiological state of cells and weakening or destroying the permeability barrier of the cell membrane [80]. This affects phytopathogens through enzymatic inhibition by damaging physiological activity and DNA alkylation, interfering with the signaling compounds of pathogenic cells and their reproductive system [81].

Different species of *Fusarium* spp. were insensitive to the fungicides copper oxychloride, Carbendaxim + Mancozeb, Hezaconazole and Valextra. Resistance generated by the fungicides is observed here [82]. Similarly, *Fusarium oxysporum* has a high risk of resistance to Fenamacril since resistant mutants were found in (800 µg mL^−1^) myosin 5 and β2-tublin [83], and different sensitivity was demonstrated with Prothioconazole and Mefentrifluconazole, obtaining higher IC_50s_ compared to other fungicides (0.22–53.52 µg mL^−1^). Fungistatic activity has been recorded with hexane extract of *J. dioica* root on *Alternaria alternata, Sclerotium rolfsii, Colletotrichum gloesporoides, Rhizoctonia solani* and *Fusarium oxysporum* with inhibition percentages of 35.9, 45.2, 19.6, 3.1 and 12%, respectively [42].

In the present study, higher growth inhibition values were obtained by both extracts (ethanol, methanol) than those reported, for example, 14.77 and 29.19% in F. *oxysporum*. This may be due to the fact that the usedsolvents have greater polarity than hexane and the greater polarity optimizes the extraction yield, trapping more free radicals and secondary metabolites [84]. Concentrated ethanolic extract of *J. dioica* root has also been reported to inhibit *F. oxysporum*, with inhibition values of 50 to 60% [85]. In this case, the inhibition percentages are higher. However, ref. [50] did not concentrate the ethanolic extract, and the variations could be due to a synergistic effect of the solvent.

Chili and corn biomass extracts obtained with deep eutectic solvents and lactic fermentation with *Lactobacillus plantarum* were evaluated against phytopathogens of economic importance for agriculture in vitro and in vivo against *Clavibacter michiganensis* subsp. *michiganensis, Xanthomonas vesicatoria, Ralstonia solanacearum, Fusarium oxysporum* f. sp. *licopersici, Colletotrichum gloeosporioides*, *Botrytis cinerea* and *Alternaria solani.* In vitro results demonstrated that the obtained extracts were effective against bacteria, unlike in vivo tests, which inhibited fungal growth [86]. Thyme (*Thymus vulgaris*) and ginger (*Zingiber officinale*) have also been reported and were evaluated against *F. oxysporum* at 4 mg mL^−1^, showing inhibition of 68.98 and 46.01% [87].

The biological activity of the secondary metabolites is reported as antioxidant, antiproliferative, hepatoprotective, antibacterial, hemolytic, thrombolytic, anti-inflammatory, antidiabetic and neuroprotective due to the presence of compounds that demonstrate pharmacological and medicinal uses [88,89,90]. For example, a 1% aqueous extract of pomegranate (*Punica granatum*) peel inhibited *F. oxysporum* mycelial growth by 40 to 45%, attributed to the total phenol content (542 mg GAE/g) and positively correlated with antifungal activity [80]. Although plant extracts with antifungal activity are limited by the development of enzyme resistance or genetic substitution processes [91], the reported activity of *J. dioica* on *B. cinerea* control in the present study represents a first report.

Differences in inhibition percentages in different phytopathogens are due to the type of plant and organ that are used, since they synthesize different compounds in greater or lesser concentrations. Some compounds are only found in specific plants or may have a higher presence of other important secondary metabolites such as cyanogenic glycosides, glucosinolates or alkaloids [80,92,93]. It is noted that plant extracts are capable of generating different biological activities, with possible uses in agriculture, such as biological control, for growth stimulation of healthy plants and as resistance inducers, decreasing disease intensity and, in response, increasing the amount of bioactive compounds [94,95].

The secondary metabolites present in plant extracts achieve antimicrobial biological activity, quantified as the inhibition or elimination of pathogenic microorganisms. However, new methods still need to be developed or adapted to further elucidate their mechanisms of action. The action has been identified as the part of the cell that interacts with secondary metabolites, such as intracellular proteins, enzymes, nucleic acids, the membrane and the cell wall. The mode of action refers to the biochemical interaction that occurs to achieve this inhibition. Both actions are evaluated through methods related to the cell wall [96].

Some studies report that phenolic acids are responsible for interfering with DNA and RNA synthesis due to their hydroxyl group, which affects the function and structure of the membrane and inactivates metabolic enzymes such as proteases, histidine, decarboxylase and amylase [97]. Phenols also alter cell permeability, causing structural changes. An altered membrane no longer performs its functions normally, such as nutrient absorption, electron transport, and protein and nucleic acid synthesis [98,99].

The most commonly used antibiotics for controlling phytopathogenic bacteria are oxytetracycline, streptomycin and kasugamycin [100]. Several crops report resistance to *Pseudomonas syringae* [66]. For example, resistance to streptomycin [101] and copper is also reported, but the disease is controlled when combined with mancozeb. However, although pesticides are an effective method for disease control, they are not a sustainable solution due to the diversity of habitats, resistance mechanisms and host distribution [102].

Other previous studies focused mainly on *J. dioica* report compounds such as diterpenes, citralitrione, 6-epi-riolozatrione, riolozatrione, jatrophatrione and jatrofolones A and B [103,104], which may be related to different biological activities of importance in the pharmaceutical field, since products can be developed with the help of nanotechnology. They can be anthelmintic [105], hypoglycemic [8], chemoprotective [106], cytotoxic [33] and antimicrobial, since they achieved 97% inhibition of *Candida albicans* at 500 µg mL^−1^ [107], and in pathogens that cause oral caries, in which its extract has a significant effect on the control of *Streptococcus mutans* [43]. Similarly, bactericidal activity has been recorded with root extract fractions on *Pseudomonas syringae pv tomato* and *Clavibacter michiganensis* subsp. *Michiganensis* with an IC_50_ of 0.5 and 1.7 mg mL^−1^ [108].

On the other hand, volumes of 10 to 50 µL of *Carum copticum* essential oil were evaluated, which obtained an inhibition halo on *Pseudomonas syringae* of 7.25 to 10 mm [109]. A dose of 1.34 mg of *Allium sativum* essential oil in 100 µL achieved an average inhibition halo of 24.66 ± 0.13, a value higher than that recorded with 1% copper sulfate with 20.66 ± 0.1 [110,111,112]. According to the literature, climatic conditions, soil type, geographic location, collection time, and even the extraction method of essential oils or plant extracts are factors that intervene in the synthesis of bioactive compounds, biological activity and how certain species are more efficient than others [113,114,115]. In this regard, several studies report the total phenol content of *J. dioica* roots and stems using different extraction methods and solvents. In the first, phenol values decreased depending on the plant organ, with the content being higher in the hydroalcoholic extract of the root compared to the stem, with 43.44 ± 0.723 µg mL^−1^ [8].

This study was evaluated in vitro, which is only an indication, since when carried out in open field or greenhouse conditions, there are many climatic variations, in which their behavior would be unknown. However, in vitro evaluation allows for a rapid response in the search for inhibition of other plant pathogen species. Therefore, further studies of *J. dioica* extract are needed in more species to identify which species are most sensitive and effective in controlling them.

The methodology and solvents required to obtain the extract are accessible for training projects with rural residents and for future *J. dioica* multiplication and extraction projects. Other plant organs can be used in cosmetics to make shampoos, infusions or soaps, which can provide additional income for residents of arid and semiarid areas of Mexico. This natural resource can be conserved through the revaluation and reorientation of its uses. Future studies can focus on the application of extracts in greenhouses and fields, identifying their mechanism of action and possible resistance, the toxicity of the extract in food, whether there are adverse effects on soil or water, assessing its incidence and severity in different crops and evaluating higher doses. There is also a history of plant extracts as growth stimulants; this would be another option for evaluation.

Combining extracts with non-biological commercial ingredients is a pending activity that could improve farmers’ perceptions of their use in the field. Alternatively, alternative (staggered) use of extracts and other ingredients could be evaluated to avoid potential antagonism, neutralization or burning of plants. In this study, the extracts were highly efficient compared to metalaxyl (metalaxyl-M: methyl N-(2,6-dimethyl phenyl)-N-(2-methoxy-acetyl)-D-alaninate at 4%, and 64% mancozeb: ethylene bisdithiocarbamate of manganese and zinc), suggesting positive prospects. As with many metabolite extraction methods, techniques can gradually improve with more sensitive equipment to determine a greater number and content of metabolites and thereby attribute their causal effect. The prospects for designing commercial formulations are encouraging for reducing environmental and health costs in pest control.

## 5. Conclusions

Ethanol and methanol extracts from the roots of *Jatropha dioica* Sessé contain phenols, flavonoids and other secondary metabolites reported to have fungistatic biological activity against *Botrytis cinerea* and *Fusarium oxysporum*, as well as bactericidal activity against *Pseudomonas syringae*. *J. dioica* is a local resource that can be used in agriculture to control pest organisms that cause food loss. The studies presented here favor the reorientation of uses and highlight the value of the species to reduce the risk of elimination from semi-desert vegetation. The solvents used are accessible and economical for farmers to obtain extracts, and future studies can focus on evaluating dosages in open field and greenhouse crops and designing commercial formulations.

## Figures and Tables

**Figure 1 microorganisms-13-01027-f001:**
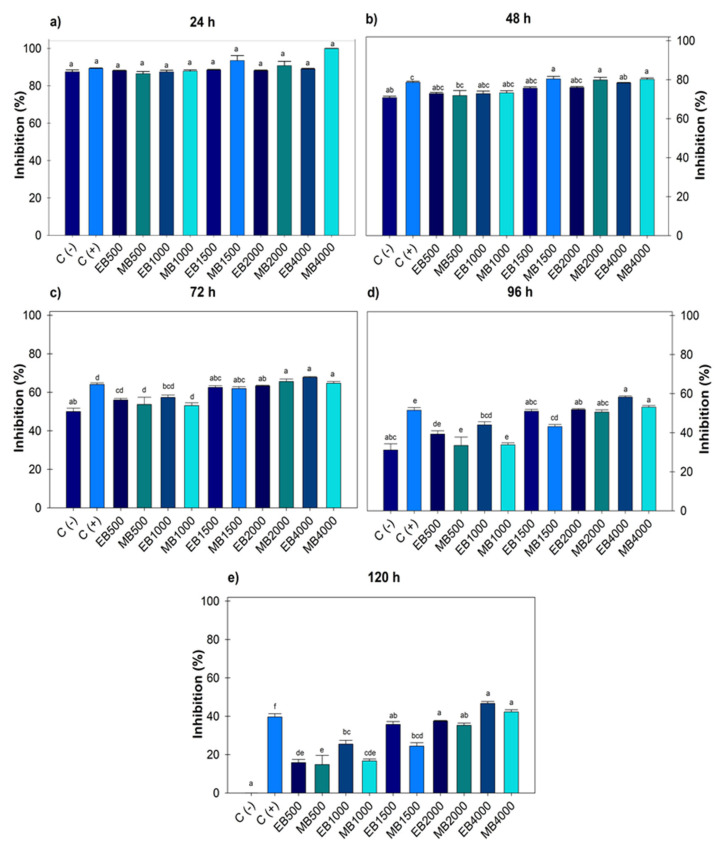
Percentage value of growth inhibition of *Botrytis cinerea* with hydroalcoholic extracts (ethanol and methanol) evaluated for five days. (**a**) evaluation at 24 h; (**b**) at 48 h; (**c**) at 72 h; (**d**) at 96 h; (**e**) at 120 h; at concentrations of 500, 1000, 1500, 2000 and 4000 µg mL^−1^. C (-): negative control. C (+): positive control with commercial fungicide Ridomil Gold. EB (*Botrytis* ethanol). MB (*Botrytis* methanol). Average values ± standard error. ANOVA, Tukey (*p* ≤ 0.05). Equal letters for each factor are significantly different.

**Figure 2 microorganisms-13-01027-f002:**
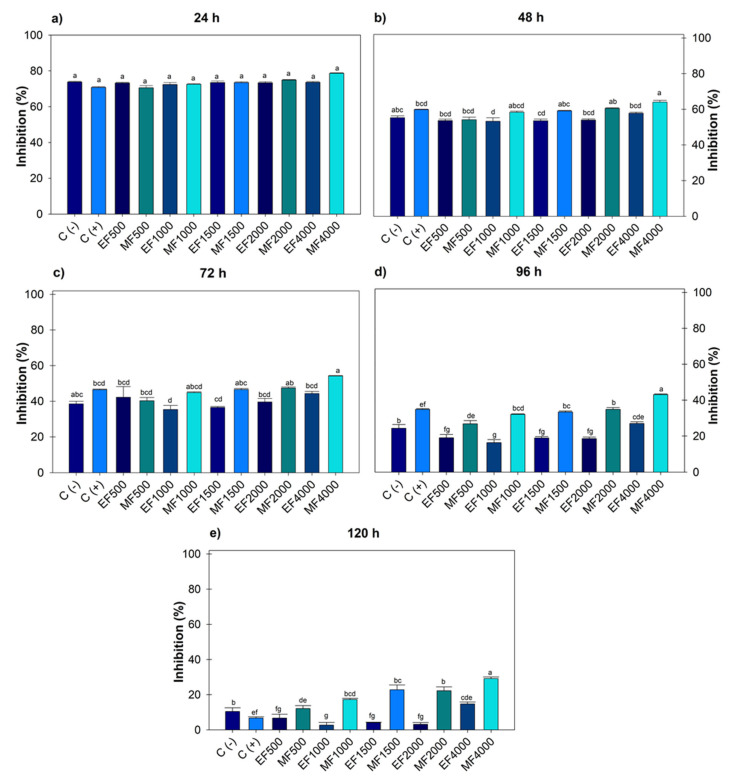
Percentage value of *Fusarium oxysporum* with the hydroalcoholic extracts evaluated for five days. (**a**) evaluation at 24 h; (**b**) at 48 h; (**c**) at 72 h; (**d**) at 96 h; (**e**) at 120 h; at concentrations of 500, 1000, 1500, 2000 and 4000 µg mL^−1^. C (-): negative control. C (+): positive control with commercial fungicide Ridomil Gold. EF (*Fusarium* ethanol). MF (*Fusarium* methanol). Average values ± standard error. ANOVA, Tukey (*p* ≤ 0.05). Equal letters for each factor are significantly different.

**Figure 3 microorganisms-13-01027-f003:**
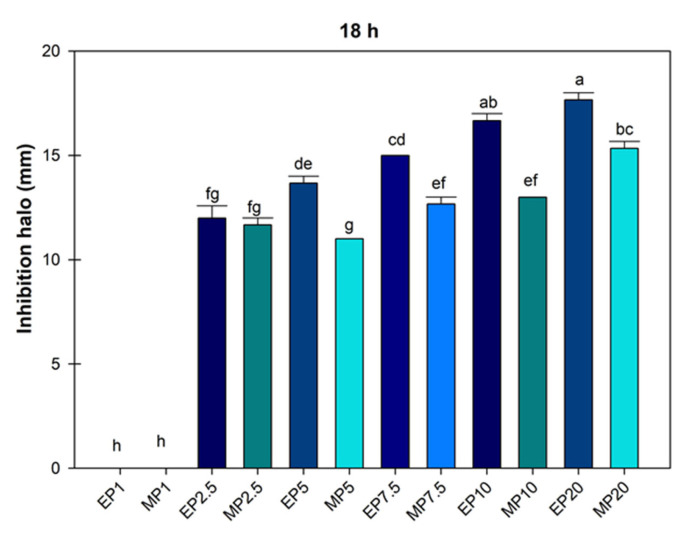
Average value of inhibition halo of *Pseudomonas Syringae* treated with hydroalcoholic extracts of *Jatropha dioica* Sessé root at doses of 1000 to 20,000 µg mL^−1^. EP1 (ethanol 1000); MP1 (methanol 1000); EP2.5 (ethanol 2500); MP2.5 (methanol 2500); EP7.5 (ethanol 7500); MP7.5 (7500); EP10 (ethanol 10,000); MP10 (methanol 10,000); EP20 (ethanol 20,000); MP20 (20,000). Average values ± standard error. ANOVA, Tukey (*p* ≤ 0.05). Equal letters for each factor are significantly different.

**Table 1 microorganisms-13-01027-t001:** Extract used for design of concentrations (µg mL^−1^) calibrated to 50 mL.

Concentration (µg mL^−1^)	Extract (mL)	H_2_O Distilled (mL)
500	5.8	44.2
1000	11.5	38.5
1500	17.3	32.7
2000	23.1	26.9
4000	46.2	3.8

**Table 2 microorganisms-13-01027-t002:** Extract (µL) used to obtain the concentrations (µg mL^−1^) calibrated to 1 mL.

Concentration (µg mL^−1^)	Extracts (µL)	H_2_O Distilled (µL)
1000	20	980
2500	50	950
5000	100	900
7500	150	850
10,000	200	800
20,000	400	600

**Table 3 microorganisms-13-01027-t003:** Total phenol (TPC) and flavonoid (TF) content of hydroalcoholic extracts of *J. dioica*. GEA: gallic acid equivalents; QE: quercetin equivalents. All values are expressed as sample dry weight.

**Solvent**	**Total Phenol Content** **(TPC) mg EAG/g**	**Flavonoid Content** **(TFC) mg QE/g**
Ethanol	8.92 ± 0.25	20.49 ± 0.33
Methanol	12.10 ± 0.34	28.21 ± 0.73

## Data Availability

Data are contained within the article.

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
