# Peer review of "Fungistatic and Bactericidal Activity of Hydroalcoholic Extracts of Root of Jatropha dioica Sessé"

_microorganisms, 2025, doi:10.3390/microorganisms13051027_

Round 1

Reviewer 1 Report

Comments and Suggestions for Authors

The article investigates the antimicrobial effect of Hydroalcoholic Extracts of Root of Jatropha dioica Sessé. It is within the scope of the journal and interesting information for phytotherapeutic approaches. However, I would highlight several points that need to be adjusted:
- Abstract: It needs to be restructured, with information on the methodology, concentrations tested, form of application and evaluation methods used. The results need to be direct, and the conclusion section concise, and directly related to the question (objective of the study).
-Introduction: Although it conveys important information, for this reviewer, it could be better contextualized by presenting the extract to be tested, with the microorganisms that were evaluated, and how the use of this substance can be beneficial in practical use. In the last paragraph, the aim study needs to be written objectively. This section encompasses many references, so I think it could be more restrictive to use, and many of them should be up-to-date (5 years).

  • Methods:
    Why do the authors use the term fungistatic and bactericidal, does this have a direct bearing on the results obtained? Or was it already expected at the beginning of the study?
    - Why the choice of such concentrations to be evaluated? Pilot study first? Reports in the literature? The concentrations seem high, was screening done with lower concentrations?
    - Tables 1 and 2 can be merged.
    - The analyses related to Total Phenol Analysis, Preparation of Gallic Acid Standard and Analysis of Flavonoids might be more interesting to present, before the fungal and bactericidal effect.
    -Results - Although it is described in the method that statistical analysis has been done, all the figures presented contain no mention of statistics. Figures 1D,1E, 2D, and 2E, it seems to me that there is a statistical difference, this is not reported.
    Figure legends need to be corrected.
    -Discussion: Although the discussion section is very long, some points are sparse. Possible mechanisms and justification for the extract's fungistatic but bactericidal effect. Explanations of the concentrations used, effects obtained, and practical use.
    What value does the analysis of Total phenol content and Flavonoid content add to the work? Why did you opt for these evaluations and what was expected in view of the evaluation of the microorganisms evaluated? This needs to be justified, or use these findings to attribute the observed effects.
    What are the limitations of the experimental model?
    What about future prospects?
    the section can be summarized.
    - Conclusion: It is important to be concise and objective, in line with the summary. Numerical results are not recommended in this section. It needs to be adjusted.

Reviewer 2 Report

Comments and Suggestions for Authors

The article "Fungistic and bactericidal activity of hydroalcoolic extracts of the root of Jatropha dioica Sessé", written by Lizeth Aguilar-Galaviz and co-authors, evaluates the antibacterial and fungicidal effects of JD extracts against phytopathogenic bacteria and fungi. The article as a whole is written and framed correctly, but some comments, questions and amendments to it are necessary.

1. There is no information on the specific identification of the pathogen species used in the study. The authors should either provide a link to the article where these strains are characterized, or the sequence number in Genbank.
2. The abstract does not clearly state the scientific novelty of the study.
3. The introduction is overloaded with well-known information, which reduces its informative value.
4. The structure of the article is not always logical: the research methods could be presented more compactly.
5. Unfortunately, the drawings are of poor quality and need to be enlarged to improve readability by potential readers.
6. The experimental methodology is not described in sufficient detail for full reproduction.
7. The description of chemical methods (for example, phenol analysis) does not contain references to standardized methods.
8. In the "Results" section, some data is presented without statistical analysis.
9. Graphs and tables are not always accompanied by sufficient analysis and interpretation.
10. The discussion lacks comparison with a wider range of similar studies.
11. Some sources are outdated or not relevant (there are references to works older than 10 years without a critical assessment of their significance).
12. The possible practical significance of the results for the agro-industrial sector has not been disclosed.
13. The conclusions contain a repetition of the information from the "Results" section without additional generalization.
14. The text contains linguistic errors and stylistic roughness that require editing.
15. The limitations of the study are not indicated, which reduces its scientific reliability.
16. There are redundant descriptions in the text that can be shortened without loss of meaning.
17. It is unclear on what medium P. syringae was grown.
18. There are inconsistencies in the work with the requirements of the journal. For example, line 99: it must be written as P. syringae.
19. Table 1, 2. Why these concentrations were chosen?
20. In the discussion section, it is necessary to write about the prospects for the development of research and how to use the results of the work in the real sector.
I urge the authors to listen to the comments and improve the quality of the article. All adjustments should be made in the text itself.

Comments on the Quality of English Language

There are linguistic errors and stylistic irregularities in the text that require editing.

Reviewer 3 Report

Comments and Suggestions for Authors

This study examines the antifungal and antibacterial properties of hydroalcoholic extracts from Jatropha dioica Sessé roots against Botrytis cinerea, Fusarium oxysporum, and Pseudomonas syringae, contributing to sustainable agriculture, biological pest control, and natural product chemistry. By exploring plant-based antimicrobial alternatives, the research addresses the need for eco-friendly solutions in plant disease management. The findings demonstrate that both ethanolic and methanolic extracts exhibit significant fungistatic activity against B. cinerea and F. oxysporum, as well as bactericidal effects against P. syringae. However, certain methodological, analytical, and experimental refinements are necessary to improve the study’s scientific rigor and real-world applicability.

One primary limitation is the lack of mechanistic validation regarding the antimicrobial action of J. dioica extracts. While the study confirms growth inhibition, it does not explore the underlying cellular effects responsible for this activity. To establish whether the extracts act through membrane disruption, enzymatic inhibition, or oxidative stress induction, additional assays would be beneficial. Conducting membrane permeability tests, such as propidium iodide staining, could help determine whether bacterial and fungal membranes are compromised. Measuring reactive oxygen species (ROS) production in treated microbial cells would provide insights into oxidative stress-related mechanisms. Additionally, discussing whether specific phenolic and flavonoid compounds interfere with microbial enzymatic pathways or quorum sensing would enhance the mechanistic understanding of the extracts’ activity.

Another critical gap is the limited investigation into the synergistic or additive effects of individual extract components. The study quantifies total phenolic and flavonoid content but does not identify which specific compounds contribute most significantly to antimicrobial activity. Performing HPLC or LC-MS profiling could help characterize the active compounds present in the extracts. Further, testing fractionated extracts to assess the antimicrobial efficacy of isolated compounds versus the whole extract would reveal whether certain compounds act synergistically. Comparing the results to known antimicrobial phenolics, such as quercetin, catechin, or chlorogenic acid, would also provide a structure-activity relationship analysis that could guide future formulation improvements.

The study would benefit from a more comprehensive statistical comparison between J. dioica extracts and commercial antimicrobial agents. While ethanolic and methanolic extracts are compared against each other, there is no direct assessment of their effectiveness relative to widely used agricultural fungicides and antibiotics. Including statistical analyses, such as ANOVA or t-tests, to compare extract efficacy with commercial products like Ridomil Gold or synthetic bactericides would help determine whether J. dioica has potential as a viable alternative or complementary treatment in plant disease management. If complete replacement is not feasible, exploring whether the extracts could reduce reliance on synthetic pesticides through combination treatments would be valuable for sustainable agricultural practices.

These questions must be answered in the revised version:

Which specific compounds in J. dioica extracts are responsible for antimicrobial activity?

Would combining J. dioica extracts with other plant-derived antimicrobials enhance efficacy?

How does J. dioica extract compare with commercial fungicides and antibiotics in terms of cost and effectiveness?

Would adjusting pH, solvent type, or extraction temperature improve bioactive compound yield?

What are the potential limitations of using J. dioica extracts in large-scale agricultural applications?

Round 2

Reviewer 1 Report

Comments and Suggestions for Authors

The points raised have been discussed, but I would like to highlight a few minor adjustments: 
- item 3.1 results is in Spanish
- the similarity rate is very high for mark #1 preprint: 60%.

Reviewer 2 Report

Comments and Suggestions for Authors

The authors have corrected most of the comments, but I still do not see the results of statistical data processing. For example, in Figures 1 and 2, the authors added information about the Tukey test, but did not indicate the differences between the variants on the graph. This is usually indicated either by an asterisk above the column or by the letter designation of the statistical group. In addition, I do not like the answers to questions in the form of a standard unsubscription. Authors should submit the document in review mode so that they can see each step of the text change, rather than the red-marked text in which the changes were allegedly made.

Reviewer 3 Report

Comments and Suggestions for Authors

/

Author Response

 Thank you for your comment.